# Employing foreign caregivers: A qualitative study of the perspectives of older stroke survivors

Yen-Nung Lin[1,2], Yosika Septi Mauludina[1], Beth E. Fields[3], Tsan-Hon Liou[2,4], Yu Su[1], Han-Ting Tsai[2], Feng-Hang Chang[1,2]*

1 Graduate Institute of Injury Prevention and Control, College of Public Health, Taipei Medical University, Taipei, Taiwan, 2 Department of Physical Medicine and Rehabilitation, Wan Fang Hospital, Taipei Medical University, Taipei, Taiwan, 3 Occupational Therapy Program, Department of Kinesiology, University of Wisconsin-Madison, Madison, WI, United States of America, 4 Department of Physical Medicine and Rehabilitation, School of Medicine, College of Medicine, Taipei Medical University, Taipei, Taiwan

* fhchang@tmu.edu.tw

**Data Availability Statement:** Data cannot be shared publicly because of the significant amount of personal and sensitive information contained in

## Abstract

### Background

Global populations are aging, and the numbers of stroke survivors is increasing. Consequently, the need for caregiver support has increased. Because of this and demographic and socioeconomic changes, foreign caregivers are increasingly in demand in many developed countries. Therefore, the perspectives of older adult care recipients regarding foreign caregivers warrants attention. This study explored the experiences of older stroke survivors receiving care from foreign caregivers in Taiwan, examining their expectations, needs, and challenges.

### Methods

This study employed a descriptive qualitative approach, conducting in-depth interviews with 23 older stroke survivors (mean age, 73.4 years; women, 47.8%). Thematic analysis was applied to transcribed data, with reflective memos aiding in meaning derivation. Methodological rigor was ensured through member checking, triangulation, and auditing.

### Results

Three major themes emerged: the motivations for hiring a foreign caregiver, expectations of stroke survivors toward foreign caregivers, and challenges related to employing foreign caregivers. Stroke survivors expected foreign caregivers to be obedient, embrace the local language and culture, and be proficient in caregiving and homemaking. Various challenges, including communication difficulties, cultural differences, skill gaps, and unfavorable attitudes and behaviors among caregivers, were noted.

the interview transcripts. The informed consent form guaranteed the privacy and confidentiality of participants' interviews, as overseen by the Office of Human Research at Taipei Medical University. Data are available from the Office of Human Research at Taipei Medical University (contact via ohr@tmu.edu.tw) for researchers who meet the criteria for access to confidential data.

**Funding:** This study was supported by Taipei Medical University – Wan Fang Hospital (grant numbers: 111-TMU-WFH-02 and 113TMU-WFH-05, PI: YL) and the Ministry of Science and Technology, Taiwan (grant numbers: MOST111-2628-B-038-015-MY3 and NSTC113-2326-B-038-002-MY3, PI: FC). The funder had no role in the study design, data collection and analysis, decision to publish, or preparation of the manuscript.

**Competing interests:** NO authors have competing interests.

## Conclusions

Foreign caregivers are a major part of the long-term care workforce and play a crucial role in stroke rehabilitation in aging Asian societies. Older stroke survivors often hire foreign caregivers to alleviate caregiving burdens, and they typically expect foreign caregivers to conform to their needs. However, employing foreign caregivers can be frustrating and stressful. Government intervention and open dialogue is necessary to improve care quality and prevent recurring caregiving problems and conflicts.

## Introduction

Stroke is a major contributor to disease burden in older adults [1]. This disabling disease imposes a direct financial burden on patients and also burdens health-care systems, social welfare systems, and labor markets [2, 3]. Family caregivers are essential for reducing this burden and assisting stroke survivors in returning to their homes and communities.

Family caregivers typically play a key role in stroke recovery [4]. In Asian cultures, caring for one's older parents is typically regarded as a family responsibility [5]. Primary caregivers of stroke survivors often experience a considerable burden due to the complexity of care [6]. Caregiving has become increasingly challenging over time due to societal changes such as more older adults working and more women working outside the home [7, 8]. These and other factors, such as more women migrating for work, often lead families in developed countries such as Singapore, China, and Taiwan to hire live-in foreign caregivers [9].

Foreign caregivers are migrant workers and are mostly young women from nearby developing countries who are employed on a full-time basis to assist older adults and individuals requiring home care [10]. In high-income countries, the demand for home-based health care has increased, and foreign caregivers have increasingly been hired to meet this demand [11]. Since 1992, the Taiwanese government has allowed hiring of live-in foreign caregivers through a regulation aimed at addressing a shortage of local caregivers for families with disabled members [12]. Taiwan has a long-term care project that offers services such as adult daycare and home care by local workers [13]; however, many individuals still prefer to hire foreign caregivers. By 2019, approximately 28% of people with disabilities were cared for by live-in foreign caregivers [14]. In 2021, the numbers of foreign and local caregivers were 225,432 and 74,601, respectively [15]. This difference is likely because foreign caregivers can be employed at a lower cost. Foreign caregivers typically earn a monthly salary of NT$18,000 to NT$21,000 (approximately US$550 to US$650), whereas domestic caregivers receive significantly higher pay, ranging from NT$70,000 to NT$80,000 (approximately US$2,300 to US$2,500) per month. [16, 17].

Formal training for foreign caregivers is minimal, and most foreign caregivers are not required to have caregiving or medical qualifications prior to their arrival in Taiwan [18]. Concerns have been raised that foreign caregivers lack the ability to provide quality care, particularly care for individuals with complex health needs, because of their lack of formal training [19].

Most studies involving foreign caregivers have focused on the challenges faced by these workers (i.e., workload and wellbeing) [20, 21]. Few studies have examined how care recipients, especially those with long-term disabilities, view their experiences with foreign caregivers. A scoping review of international literature highlighted the lack of research in Asia on care

recipients' experiences with foreign caregivers [21]. Notably, few studies have explored how individuals with care needs and their families perceive foreign caregivers. Several qualitative studies have explored the complex relationships between foreign caregivers and recipients [21–23]; however, no studies have specifically focused on the needs and expectations of stroke survivors toward foreign caregivers. Given Taiwan's aging society and the growing number of stroke survivors facing unique challenges [24], understanding their caregiving experiences is crucial. The present qualitative study described the experiences of older stroke survivors in Taiwan who employ foreign caregivers, focusing on their expectations, needs, and challenges. The findings will help improve care services, assist policymakers in identifying unmet needs, and enhance care quality, which can support caregivers and guide policy development for long-term care systems around the world.

## Methods

### Design

This study employed a descriptive qualitative approach [25–27] to understand stroke survivors' experiences in receiving care from foreign live-in caregivers. The study was reviewed and approved by the Institutional Review Board of the affiliated university and all participating research sites (approval number: N201907057). Informed consent was obtained from all participants before interviews were conducted, and participants were informed they had a right to withdraw from the study at any time. The Consolidated Criteria for Reporting Qualitative Research checklist [28] was used to evaluate this study's methodological reporting quality (**S1 File**).

### Study setting and recruitment

Participants were recruited from the rehabilitation outpatient clinics of three academic medical centers in greater Taipei, Taiwan. Taiwan's National Health Insurance places no restrictions on the duration of outpatient rehabilitation services for stroke patients, allowing them to access these services for as long as needed. This policy enabled us to recruit patients at different stages of recovery. Patients with stroke were screened for study inclusion eligibility by staff at the rehabilitation clinics, and potentially eligible patients were introduced to the study researchers, who then met with the patients in person and explained the purpose of the study by using informational leaflets. Recruitment continued on a rolling basis until data saturation was reached [29].

### Study participants

The criterion sampling approach [30] was adopted to identify potential participants. Patients were included if they had received a diagnosis of stroke, were aged ≥60 years, were enrolled in an outpatient rehabilitation program, had been receiving 24-hour home care from a live-in foreign caregiver for at least 6 months, could communicate in Mandarin, could engage in an interview for at least 60 min, and could understand and provide informed consent. Patients were excluded if they had severe aphasia (determined using the aphasia subscale of the National Institutes of Health Stroke Scale [31]) or a medical complication that might have prevented them from participating in interviews. Written informed consent was obtained from all participants between December 3, 2020, and October 20, 2021.

### Positionality of the research team

The research team comprised individuals from diverse professional backgrounds. Each professional contributed unique expertise to the study. YL and TL, who were both male professors

and attending physicians at a teaching hospital in Taiwan, have extensive experience in stroke rehabilitation. YM, a female Indonesian PhD student, specializes in geriatric physical therapy and qualitative research. BF, a female associate professor in an occupational therapy department in the United States, is an occupational therapist with expertise in caregiving, aging, and geriatric health. YS, a female research assistant from Taiwan, has a master's degree in injury prevention and is trained in qualitative research. HT is a male occupational therapist at a teaching hospital in Taiwan. FC, the senior researcher, is a female professor in Taiwan with expertise in rehabilitation sciences. Both BF and FC have significant experience in qualitative research and led the study.

To minimize personal biases, YS and YM, who had no prior relationships with the participants, managed recruitment. YS conducted all interviews in Mandarin. FC, YM, and YS led data analysis, with regular team meetings held to reflect on how each researcher's professional background might affect data interpretation. These discussions helped address the potential influence of personal assumptions, ensuring the integrity of the findings.

## Procedure

Semistructured interviews were conducted to obtain in-depth insights from the participants. An interview guide comprising seven open-ended questions was developed and reviewed by an interdisciplinary expert panel, stroke survivors, and caregivers. The full interview guide is available in **S1 Table**. Three of the seven questions are as follows:

1. What led you to decide to hire a foreign caregiver?

2. What do you believe are the duties and responsibilities of foreign caregivers?

3. What benefits or challenges have you experienced after employing a foreign caregiver?

Interviews were conducted between December 2020 and October 2021. The interviews were scheduled at times that were convenient for the participants and took place in quiet settings, either at academic medical centers or in the participants' homes. Family caregivers were encouraged to accompany the participants and provide assistance as necessary to answer the questions. The presence of caregivers was solely for assistance purposes, and if any interference in the interview process was observed, the caregivers were asked to leave to minimize their influence on the interview.

Each participant was interviewed once, and each interview lasted 60–90 min. The interviews were audio-recorded and transcribed verbatim by the interviewer. One of the other researchers assessed the accuracy of the transcripts.

Participant demographic and clinical information was collected from medical charts and by using a questionnaire. Data were handled confidentially. Participant information was deidentified by using unique identification numbers that were stored separately from the study data on a secure, password-protected drive. Only the principal investigator and select research staff had access to the encrypted file linking identification numbers to participants. The Office of Human Research was responsible for monitoring and auditing the collected data to ensure compliance with ethical standards.

## Data analysis

All data were stored, managed, and analyzed using NVivo 12 Pro (QSR International, Melbourne, Australia). Thematic analysis was performed to interpret the meaning of textual data [32]. Two researchers independently read, abstracted, and coded the transcripts immediately after each interview and then translated the coded data and quotes from Mandarin to English.

Translation accuracy was verified by a third researcher. Data with similar meanings were grouped and synthesized into descriptive themes, which were reviewed by the entire research team [33]. For example, quotes that were coded as "obedient" and "respectful" were grouped under the subtheme "Being obedient and submissive." Analysis continued until thematic saturation was reached, with co-coders agreeing that no new categories emerged [33, 34]. Translated coding results were reviewed and discussed until consensus was reached on the final interpretation of participants' experiences [34, 35].

Between-methods triangulation [36] was used to enhance the credibility of the findings by cross-verifying data collected from different methods and sources. Interview data were compared with the interviewer's observations and reflective memos. YS and YM recorded these comparisons during the interview process to better understand the meaning of the data. Additionally, we conducted member checking, where YS shared the analysis results with the participants in person or through phone calls to confirm the accuracy of the data interpretation [37].

## Results

A total of 28 stroke survivors were invited to participate in the study. Of these, 23 completed the interviews, 3 withdrew due to scheduling conflicts, and 2 declined participation. Data saturation was reached after interviewing 20 participants, with 3 more interviews conducted to ensure no new themes emerged. Recruitment concluded after 23 participants were interviewed. The participant demographics are presented in Table 1. Overall, 47.8% of the participants were women, the average age was 73.4 years, the average time since stroke was 3.5 years, and the average Barthel Index score was 51.

We identified 3 major themes with 11 subthemes. Quotes supporting each theme are provided in the following sections.

### Theme 1. Motivations for hiring a foreign caregiver

Several reasons were provided for hiring foreign caregivers. The reasons included needing supervision and physical assistance, lacking family caregiving support, and cost.

**Need for supervision and physical assistance.** Foreign caregivers were critical in helping the participants perform daily living activities. Caregivers also ensured the safety of the participants. One participant said, *"I can't walk without her. I need her to be with me at all times."* (PS01)

Another participant said, *"After I had a second stroke, I decided I needed someone to stay with me to ensure my safety."* (PS13)

**Lack of family caregiving support.** The primary reason for hiring a foreign over a local caregiver was a lack of family caregiving support. Some participants lived alone, without family nearby. Some participants lived with family members who did not have time to take care of them. *"I live alone. My husband has passed away. My son lives nearby, but he doesn't have time to take care of me."* (PS12)

Some participants lived with family members who were available to provide care but unable or unwilling to be a primary caregiver.

*"My family basically can't do anything about caregiving. They are not very busy either. They just want to live their own lives. We discussed with hospital staff whether my family would be able to assist me, and the doctor advised me to hire a caregiver."* (PS21)

**Table 1. Demographic data.**

| Participant ID | Age (years) | Sex | Education level | Monthly household income (New Taiwan Dollar) | Religion | Primary caregiver | Time since stroke (months) | Barthel Index score (out of 100) | Duration of foreign caregiver employment (months) | Self-reported relationship with foreign caregivers |
|---|---|---|---|---|---|---|---|---|---|---|
| PS01 | 85 | Female | Junior high school | 0–29,999 | Buddhism | Foreign caregiver | 20 | 70 | 13 | Very good |
| PS02 | 78 | Female | Elementary school | 30,000–49,999 | Buddhism | Foreign caregiver | 60 | 55 | 50 | Not good |
| PS03 | 80 | Female | Elementary school | 30,000–49,999 | Buddhism | Foreign caregiver | 96 | 50 | 84 | Very good |
| PS04 | 71 | Male | Graduate school | 80,000–119,999 | Buddhism | Foreign caregiver and spouse | 39 | 15 | 30 | Very good |
| PS05 | 63 | Male | High school | 0–29,999 | Buddhism | Foreign caregiver | 4.5 | 60 | 12 | Not good |
| PS06 | 88 | Male | College degree | 50,000–79,999 | No | Foreign caregiver and children | 6 | 95 | 18 | Very good |
| PS07 | 61 | Male | College degree | 0–29,999 | Christian | Foreign caregiver and spouse | 15.5 | 50 | 9 | Good |
| PS08 | 63 | Male | High school | >120,000 | Buddhism | Foreign caregiver | 12 | 55 | 12 | Very good |
| PS09 | 81 | Female | Elementary school | 0–29,999 | Buddhism | Foreign caregiver | 33 | 70 | 26 | Good |
| PS10 | 63 | Male | High school | 30,000–49,999 | Buddhism | Foreign caregiver | 40 | 85 | 30 | Good |
| PS11 | 68 | Female | High school | 30,000–49,999 | Buddhism | Foreign caregiver | 13 | 50 | 8 | Not good |
| PS12 | 68 | Male | College degree | 50,000–79,999 | Christian | Foreign caregiver | 50 | 75 | 30 | Good |
| PS13 | 81 | Male | Junior high school | 30,000–49,999 | Buddhism | Foreign caregiver | 25 | 0 | 20 | Good |
| PS14 | 84 | Female | Elementary school | >120,000 | No | Foreign caregiver | 25 | 35 | 24 | Good |
| PS15 | 69 | Male | College degree | 30,000–49,999 | Christian | Foreign caregiver and nurse | 108 | 0 | 96 | Good |
| PS16 | 78 | Male | High school | 30,000–49,999 | Buddhism | Foreign caregiver | 19 | 70 | 16 | Good |
| PS17 | 70 | Female | Elementary school | 0–29,999 | Buddhism | Foreign caregiver | 120 | 35 | 24 | Good |
| PS18 | 70 | Male | College degree | 30,000–49,999 | Christian | Foreign caregiver | 20 | 95 | 20 | Very good |
| PS19 | 69 | Female | Elementary school | 30,000–49,999 | Buddhism | Foreign caregiver | 36 | 65 | 22 | Very good |
| PS20 | 80 | Female | Elementary school | 50,000–79,999 | Buddhism | Foreign caregiver and spouse | 36 | 0 | 30 | Good |
| PS21 | 69 | Female | High school | 30,000–49,999 | Buddhism | Foreign caregiver | 24 | 75 | 12 | Okay or moderately good |
| PS22 | 84 | Female | Elementary school | 50,000–79,999 | Buddhism | Foreign caregiver | 160 | 20 | 70 | Good |
| PS23 | 65 | Male | College degree | 30,000–49,999 | Buddhism | Foreign caregiver | 20 | 50 | 36 | Very good |

**Financial limitations.** Several participants explained that they hired foreign caregivers instead of local caregivers because of cost considerations. One participant said, *"Local caregivers are too expensive. They cost NT$1,000 to NT$2,000 per hour."* (PS15)

Another participant said *"The cost of hiring a foreign caregiver is around $NT26,000 per month. This is much cheaper than hiring a local caregiver, which costs around NT$75,000 per month."* (memos recorded by YS, March 2021)

Participants selected foreign caregivers over local caregivers because hiring a foreigner caregiver is half as expensive. *"If we could hire a Taiwanese caregiver, that would be ideal because then we wouldn't have to deal with language and culture barriers. But Taiwanese caregivers are just too expensive. My sons can't afford it."* (PS27)

## Theme 2. Expectations of participants toward foreign caregivers

The participants held various expectations for their foreign caregivers. The participants expected caregivers to be obedient and submissive, to learn Mandarin, to embrace Taiwanese culture, to master stroke care and homemaking, and to provide care for other family members.

**Being obedient and submissive.** Most participants expected their caregivers to be obedient and submissive. Stroke survivors and their families tend to see themselves as masters with an ownership claim on their foreign caregivers. They expect caregivers to "listen to the master" and complete all assigned tasks, including providing care, performing household chores, and handling personal and family responsibilities. This expectation stems from the fact that the caregivers are paid and employed to work for them 24 hours a day. *"She must do everything we tell her to do. That's her job, plain and simple."* (PS01)

*"I'm the one paying, so she better listens to me, right? Otherwise, what's the point of having her here?"* (PS06)

**Learning mandarin and embracing taiwanese culture.** Nearly all participants expected their foreign caregivers to speak their language (Mandarin or Taiwanese) and adapt to Taiwanese culture. Several participants contended that effective communication with their foreign caregivers is only possible if they speak Chinese or Taiwanese. *"If she doesn't understand Chinese, how can she know what I'm saying?"* (PS03)

The participants also expected their caregivers to embrace Taiwanese culture, the Taiwanese diet, Taiwanese religious practices, and Taiwanese societal rules. The participants believed their interactions with their caregivers would be smoother if the caregivers integrated into Taiwanese culture.

*"She's in Taiwan now, so she should just go with the flow and adapt to Taiwanese customs, like saying 'please,' 'thank you,' and 'excuse me.' That's how we Taiwanese talk."* (PS06)

**Mastering stroke care and homemaking.** All of the participants believed that foreign caregivers should have caregiving experience and be capable of caring for stroke survivors. The participants with functional limitations expected their caregivers to assist them with transferring, eating, showering, toileting, and dressing.

*"My caregiver should be at least capable of helping me get in and out of bed, shifting my position, and changing my diaper."* (PS17)

The participants and their families believed that caregivers should do housecleaning, do household chores, do laundry, run errands, and prepare meals. *"When she's not busy taking care of me, she should be cleaning the house, cooking, taking out the trash—stuff like that. After all, we paid her to work for us 24 hours a day. She should not be sitting there doing nothing."* (PS02).

*"Sometimes we ask her to buy fruits and vegetables or to go to the supermarket to buy milk."* (PS22)

**Providing care for other family members.**   Foreign caregivers are expected to care for other family members in addition to their primary care recipient. For example, caregivers may be expected to babysit children, take care of other older family members, and cook for the entire family. The participants considered these additional tasks to be part of the caregiver's job. One participant said, *"If she doesn't do it, then no one will."* Another participant said, *"I expect her to cook dinner for our whole family, so when everyone gets home from work, we can all eat together."* (PS21).

Another participant said, *"I also need her to help bathe my son. He's 8 years old."* (PS05)

## Theme 3. Challenges related to employing a foreign caregiver

Several participants provided examples of the challenges they encountered when engaging with their foreign caregivers. The challenges were related to communication difficulties, cultural differences, caregiving skill deficiencies, and problematic attitudes and behaviors. These challenges caused substantial frustration and distress for the participants.

**Communication difficulties.**   Several participants were unable to easily communicate with their caregivers, and communication difficulties often led to misunderstandings. One participant said, *"I talk to her; she doesn't understand a word. I ask her to hold my hand, and she still doesn't get it—it's like talking to a brick wall! It just makes me so angry!"* (PS02). Because of these communication challenges, participants often felt that their caregivers struggled to perform certain tasks effectively, such as running errands or following medical instructions. These difficulties significantly impeded the caregiving process.

**Cultural differences.**   Many participants did not initially expect that cultural differences would be a concern. However, many reported dietary and religious differences that they had noted. For example, most Indonesian people do not eat pork, which is popular in Taiwan, and most Indonesian people are Muslim, whereas most Taiwanese people are Buddhist or Taoist. Various differences in behaviors, habits, and customs between the participants and their caregivers resulted in conflict. *"Due to cultural differences, their gestures and actions can be more pronounced and a bit rough. Although there's no ill intent, it can sometimes be startling."* (PS25).

*"She is Muslim and does not eat pork. This causes a bit of trouble for us. When we cook together, we have to avoid pork. This is a bit of a hassle."* (PS26).

*"She prays several times a day, every day. She refuses to help while praying. For example, she won't help with things like clearing phlegm or changing diapers while she is praying."* (PS27)

Because of these differences, the participants experienced considerable frustration during the caregiving process and were required to continually adjust their thoughts and behaviors and changed their deeply ingrained beliefs.

**Caregiving skill deficiencies.** Many participants were disappointed with their foreign caregivers because of their lack of knowledge and skills in caring for stroke survivors. This disappointment is often heightened for first-time foreign caregivers who have no experience caring for older adults with disabilities and likely have not received training. *"My caregiver didn't have any experience with taking care of patients or older people before she came to our house."* (PS04)

*"My caregiver had zero caregiving training. She even caused me to fall over when she first arrived."* (PS07)

Some participants complained that their caregivers lacked household skills, citing poor cooking, ineffective cleaning, or an inability to maintain a tidy environment. *"She is completely inept at housework, and cooking is out of the question. The things she cooks are inedible."* (PS24)

**Concerns with foreign caregivers' attitudes and behaviors.** Nearly all participants were disappointed with their foreign caregivers' attitudes and behaviors, citing issues such as a lack of obedience, a lack of focus, a lack of productivity, and even engagement in unethical conduct.

One participant shared an example of a foreign caregiver who frequently refused her requests. The caregiver often refused to help the participant with dressing, toileting, and going on outings. The caregiver also complained that the participant was difficult to care for.

*"She helped me pull up my pants, but it was crooked. I told her it felt uncomfortable. I told her the pants were not pulled all the way up. I asked her to help me pull them up properly, but she got mad and called me difficult."* (PS01)

Participants were distressed and disappointed because their foreign caregivers could not focus. The most common complaint among participants was that foreign caregivers were constantly on personal phone calls, from morning till night, even during work hours, disregarding their duties. This deeply bothered nearly all of the participants, who were with their foreign caregivers 24 hours a day. *"She's on the phone all the time, from morning till night. She talks on the phone even while helping me bathe, totally distracted. Pushing my wheelchair while chatting? Super dangerous. What if I bump into something?"* (PS07)

*"She's on the phone all day, literally nonstop. Chatting with friends, chatting with family, never a moment without phone in hand. She's totally unfocused on her tasks. And she talks so loudly on the phone, it's so annoying!"* (PS14)

*"She's always on the phone with her friends or family in Indonesia, talking in a language I don't understand, and it's so loud. I'm actually not a fan of loud noises, and hearing her talking all day drives me crazy. Trying to talk to her about it doesn't work—she just keeps on talking all day. Sigh."* (PS18)

Some participants even reported that their caregivers engaged in unethical behavior. For example, some caregivers would ignore them, leave them alone, steal money, or run away. *"She's always on the phone, and once she drops me off at the rehab room, she disappears. Nurses have to help me find her. She's here one moment, and gone the next, vanishing for half an hour."* (PS09) *"I left my wallet on the table with NT\$20,000 inside. I went into the room for just a moment, and when I came out, NT\$10,000 was missing. Only the two of us were home. Who do you think took it?"* (PS02)

*"She got her salary, said she had to do something downstairs, and then never came back. She just ran off."* (PS12)

Additional sample quotes supporting the themes and subthemes are provided in **S2 Table**.

## Discussion

As the world shifts toward becoming an aged society, many developed countries have introduced migrant workers into their long-term care systems to shift the burden of care from families to society [22, 38, 39]. Although the reasons for recruiting foreign caregivers to meet long-term care needs might be similar across these countries, care recipients' experiences can vary considerably because of differences in societal backgrounds and cultural contexts. This study explored the perspectives and experiences of stroke survivors in Taiwan concerning foreign caregivers. In line with the global trend of foreign caregivers becoming a crucial part of the long-term care workforce in many developed countries because of their changing socioeconomic, cultural, and familial structures [40], our findings suggest that the primary reason stroke survivors hire caregivers for long-term care is because family members are unable to provide this care. Furthermore, the primary reason stroke survivors choose to hire foreign caregivers instead of local caregivers is that foreign caregivers are less expensive to hire. In Taiwan, individuals with disabilities or severe chronic illnesses may employ foreign live-in caregivers [41]. Stroke survivors, who are qualified to apply for foreign caregivers, are often advised by clinicians to seek such support to assist with daily living following hospital discharge. Hiring live-in foreign caregivers enables these older people to live at home with their family instead of at an institution [12]. This is also a primary reason foreign caregivers have become a notable source of formal long-term care labor in aging Asian societies [42].

The stroke survivors who participated in interviews in this study shared their expectations for their foreign caregivers. Caregivers were expected to be obedient, to embrace the local language and culture, to master caregiving and homemaking, and to care for other family members. Nearly all of the participants hired foreign caregivers to assist with household chores in addition to providing care. Foreign caregivers were expected to set aside their language, culture, and ideas and dedicate themselves entirely to the employer and the employer's family, with a commitment to providing 24-hour services. This raises concerns about fairness and respect for the caregivers' autonomy. The participants often overlooked their caregivers' own cultural values, rights, and limitations. They developed a strong and unequal hierarchical dynamic that resembles a master–servant dynamic rather than a mutually respectful employer–employee relationship. Employers and agents in Asian societies often refer to foreign caregivers as maids, treating them as the property of their employees rather than as human beings with their own will and freedom. This belief is particularly prevalent in Sinitic cultures, in which a master is considered to have control over their maid's body and personhood [42].

The reality of employing a foreign caregiver did not fully align with the expectations of stroke survivors and their families. Unexpected challenges—such as communication difficulties, cultural differences, caregiving skill deficiencies, and improper caregiver attitudes and behaviors—frequently left stroke survivors struggling to cope and sometimes caused considerable distress. These challenges reflect the fact that stroke survivors and foreign caregivers tend to have different expectations regarding the role that foreign caregivers should play. Stroke survivors believe that foreign caregivers should obey their employers; relinquish their original language, culture, and personal life; and know how to provide care and perform household duties. Research indicates that foreign caregivers do not wish to be treated as property or as

servants. They believe that they should be provided with private time and space to connect with their family and friends and a reasonable amount of time off for rest and recuperation [20]. Differences in expectations can create tension between stroke survivors and foreign caregivers, potentially affecting quality of care and the health and wellbeing of both parties [42, 43]. Differences in perceptions between stroke survivors and foreign caregivers stem from a range of complex factors, including culture, social class, and institutional systems [42]. Government attention and active dialogue is necessary to identify solutions to the aforementioned problems to prevent deterioration of the relationships between stroke survivors and their caregivers.

Another concern that emerged from the findings of the current study is that many stroke survivors discovered that their foreign caregivers had not received formal training before starting work, leading to various problems, such as a lack of knowledge and skill in caring for the stroke survivors, communication difficulties, and unprofessional attitudes and behaviors. Foreign caregivers tend to receive little or no training before starting their caregiving jobs [12], and most foreign caregivers are not required to have caregiving or medical training before arriving in Taiwan, although some agencies provide brief training sessions to foreign caregivers upon their arrival in Taiwan [18]. This lack of formal training has raised concerns about quality of care, especially for patients with more complex health needs [19]. Many foreign caregivers are not adequately prepared to provide professional care or effectively communicate with their care recipients [12, 20]. Notably, among foreign caregivers interviewed in another study, the primary reason for choosing to work overseas as a caregiver was the minimal job training requirements [20]. Training is essential for safety. One participant in the current study recounted an incident in which his caregiver's inexperience caused him to fall early in their working relationship. Some participants also reported feeling neglected or disrespected. Others did not receive adequate attention from their foreign caregivers. These problems require government attention and proactive intervention. Foreign caregivers must receive training before and during employment to effectively enhance care quality and prevent recurring caregiving problems and conflicts.

This study has several strengths. First, this study is the first qualitative exploration of the perspectives of stroke survivors in Taiwan regarding their experiences with foreign caregivers. The use of in-depth, face-to-face interviews allowed for the collection of rich, detailed data. The study's methodological rigor, including its use of triangulation, member checking, and reflective memos, further enhanced the trustworthiness and credibility of the findings.

This study has several limitations. Our participants were older stroke survivors recruited from three medical centers in the greater Taipei area; the results may not be generalizable to individuals in other areas. Future studies should consider including participants from other regions to gain a broader understanding. Moreover, our participants were stroke survivors, and their cognitive functions and language abilities may have affected the quantity and quality of information they shared. Future studies could employ observational methods and questionnaire surveys to address these biases when collecting data from care recipients and their families.

## Conclusions

In Taiwan, stroke survivors often hire foreign caregivers to alleviate the burden of caregiving on their family. However, stroke survivors tend to see foreign caregivers as servants and expect caregivers to conform to their needs. Communication difficulties, cultural differences, and unfavorable attitudes and behaviors among caregivers are concerns that cause stroke survivors frustration and distress, and stroke survivors often believe that foreign caregivers require more

training. Government intervention is necessary to address these challenges and ensure foreign caregivers are able to perform their roles effectively. The government should provide training and support to foreign caregivers, bolster caregiving services, and foster more balanced and equitable relationships between caregivers and care recipients. Hiring families should have realistic expectations about the roles, capabilities, and limitations of foreign caregivers. A system of ongoing cultural competency training for both employers and caregivers could help minimize misunderstandings and foster mutual respect. Additionally, clear ethical guidelines could be established to protect the rights of foreign caregivers while ensuring that stroke survivors receive the quality care they need.

## Supporting information

**S1 File. Consolidated Criteria for Reporting Qualitative Research (COREQ) checklist.**
(DOCX)

**S1 Table. Interview guide.**
(DOCX)

**S2 Table. Sample quote.**
(DOCX)

## Acknowledgments

We thank the staff members at the collaborating hospitals, that is, Taipei Medical University Hospital, Taipei Municipal Wanfang Hospital, and Shuang Ho Hospital. We extend particular thanks to the collaborating therapists, including Yen-Ting Liu, OTR/L (Taipei Municipal Wan Fang Hospital), and Jui-Chi Lin, OTR/L (Shuang Ho Hospital).

## Author Contributions

**Conceptualization:** Feng-Hang Chang.

**Data curation:** Tsan-Hon Liou, Yu Su, Han-Ting Tsai.

**Formal analysis:** Yosika Septi Mauludina, Yu Su.

**Funding acquisition:** Yen-Nung Lin, Feng-Hang Chang.

**Investigation:** Yosika Septi Mauludina, Tsan-Hon Liou, Yu Su, Feng-Hang Chang.

**Methodology:** Beth E. Fields.

**Project administration:** Yen-Nung Lin, Tsan-Hon Liou, Feng-Hang Chang.

**Resources:** Beth E. Fields, Tsan-Hon Liou.

**Supervision:** Yen-Nung Lin, Beth E. Fields, Feng-Hang Chang.

**Validation:** Yosika Septi Mauludina, Beth E. Fields, Yu Su, Han-Ting Tsai.

**Writing – original draft:** Feng-Hang Chang.

**Writing – review & editing:** Yen-Nung Lin, Yosika Septi Mauludina, Beth E. Fields, Tsan-Hon Liou, Yu Su, Han-Ting Tsai.

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
