## [Decision Letter · Decision Letter 0]

24 Sep 2024

PONE-D-24-31334Employing foreign caregivers: exploring perspectives of older stroke survivors - a qualitative studyPLOS ONE

Dear Dr. Chang,

Thank you for submitting your manuscript to PLOS ONE. After careful consideration, we feel that it has merit but does not fully meet PLOS ONE’s publication criteria as it currently stands. Therefore, we invite you to submit a revised version of the manuscript that addresses the points raised during the review process.

**ACADEMIC EDITOR: **

In view of difficulty in securing reviewers, some of whom did not revert after accepting the review, I have taken up the role as one of the reviewers for this manuscript.

Both reviewers have raised pertinent points that require attention. Please address all the comments provided by the reviewers carefully and ensure that all points are addressed in the revision. 

As the authors have claimed that they adhere to COREQ checklist in the manuscript, please upload a completed COREQ checklist to ensure that hte criteria have been met. 

We look forward to receiving your revised manuscript.

Kind regards,

Chai-Eng Tan

Academic Editor

PLOS ONE

Journal Requirements:

"NO authors have competing interests"

Reviewers' comments:

Reviewer's Responses to Questions

**Comments to the Author**

1. Is the manuscript technically sound, and do the data support the conclusions?

Reviewer #1: Yes

Reviewer #2: No

2. Has the statistical analysis been performed appropriately and rigorously? 

Reviewer #1: N/A

Reviewer #2: N/A

3. Have the authors made all data underlying the findings in their manuscript fully available?

Reviewer #1: No

Reviewer #2: No

4. Is the manuscript presented in an intelligible fashion and written in standard English?

Reviewer #1: Yes

Reviewer #2: No

5. Review Comments to the Author

Reviewer #1: Review ”Employing foreign caregivers: exploring perspectives of older stroke survivors – a qualitative study”

Thank you for the opportunity to review this manuscript with, from my point of view, a problematic topic. The authors themselves mention exploitation and I become very badly affected when reading the results. The expectations of the persons affected by stroke widely exceeds what is reasonable to expect. How can anyone consider it to be ok to expect persons to work 24 hours per day as is stated in line 250? This represent a very dehumanized way of looking at other persons. The researchers make a good discussion of this very problematic issue.

Abstract

In the section for Results, the themes are given, however, not with the correct names. Make sure to use the correct names from the results or re-write the text in the abstract so it is evident that what is written there is not the actual themes, but a summary of what they tell about.

Introduction

A little more about the health care system in Taiwan needs to be described for an international audience to be able to understand the article. Which care services are available for persons who are in need of daily care, but not needing hospital care? What is the cost for such services for the individual person in need?

On p 11, lines 86-87, it is stated that “a” scoping review has highlighted but then two references are given. When using two, the word “a” is not sufficient. However, reference 16 does not seem to be a scooping review.

On p 11, lines 87-90, a statement is given about existing qualitative investigations, where the scoping review is given as one of the references. Are the authors sure that all studies included in the scoping review are performed with qualitative methods?

Method

Setting and participants

It is unclear to me if all patients with a previous diagnosis of stroke are scheduled for outpatient visits at the physical medicine and rehabilitation departments of the included hospitals or if the patients who are scheduled for such visits constitute a special selection in some way? I would not expect that everyone who has got a diagnosis of stroke and live in the catchment area of one of the included hospitals, would still be scheduled for outpatient visits at these departments more than five years after their stroke. This needs to be clarified in order to understand if the available participants in any way constitutes a selected group of patients.

Procedure

On p. 13 the development of the semi-structured interview guide is described. I would suggest to also give the number of questions, not only state that there were “various questions” together with the examples tat are given.

On p 14, line 140, present tense is used, while past tense should be used, on for example “family caregivers are encouraged”.

Data analysis

On p 12, line 102 the study is stated to employ a descriptive phenomenological approach, but in the section for Data analysis, p 14, line 149, it is stated that a thematic analysis was performed. These two statements are not compatible and cannot both be true Please revise this. I cannot find any signs in the text, of this study being a phenomenological study so I would suggest to keep the claim about the thematic analysis and omit the phenomenological claims. This also gives a need to revise the aim. When a phenomenological analysis has not been performed, the results is not “lived experience”.

Ethical considerations

I cannot find any information concerning ethics despite the statement about ethical approval and information given to the participants found on p 12. Information about if informed consent was obtained, how data was handled and stored and how the participants’ identities have been protected are missing. Also information is lacking about the relationship between the researchers who recruited the participants and the researcher who conducted the interviews, and the participants. Were the researchers in any caregiving position for the participants? Can it be assumed that any power relations between the researchers and the participants existed so that the participants could feel forced to participate?

Results

In the method section only patients with a diagnosis of stroke are stated to be participants. I was therefore very surprised to understand from line 181-182 and the very first quotation in the results, that information given by relatives to the participants was included. If relative’s statements are to be part of data, then they also needs to be viewed as participants in the study.

There are multiple problems related to the results, mostly pertaining to the headings of the themes and sub themes.

The sub theme “Severity and complexity”, I do not see nether severity nor complexity in the text that describes what is labelled as this. Please revise.

I am not very sure about the name of the second theme, line 208. Is the use of “survivor” really good? We can assume that no dead person will have the need of a caregiver of any kind. I suggest to use the word participant or the word stroke instead.

I have trouble to understand how anyone hiring a foreign caregiver can view language barrier as an “unexpected consequence”. Is really seen in the interviews or is this a result of a bad heading for the sub theme?

Make sure that this section reports on the results, not discuss them. Discussion has a section dedicated to that. Examples that needs to be removed from the results are for example line 225.

It is not ok to use valuing word in the results section, these should be reserved for the discussion, see for example first word, line 309.

Discussion

Well-written and problematizing the results in a good way. This is really well needed with this kind of problematic and bothering results showing so little respect for other human beings.

Strength and limitations

Discusses the limitations in a proper way. No strengths in the study is reported which I consider would be of value. There cannot only be limitations in the study, there must also exist some good things.

I also do not agree that there would be greater reasons for persons with negative experiences to decline participation, than it would be for persons with positive experiences. Those who are dissatisfied are often not hesitant to tell about it.

Conclusion

Supported by the results.

References

Reference 19 – the title of the publication have been written in a mysterious way: Approaches QIaRDCAF. Similar problem appear in ref 33.

The journal names are alternately written with initial upper-case and lower-case in each word. Make sure to adhere to the journals guidelines concerning this.

Language

Good

Reviewer #2: Thank you for addressing an important phenomenon in caregiving for stroke patients in an Asian country. In Asian countries, reliance on informal caregivers is common due to the reluctance to institutionalise stroke patients in care homes, related to their local culture of filial piety. The current manuscript still contains many areas that require improvement.

I hope the authors can address the following issues:

The study is being presented as a phenomenological study. Please mention the key characteristics of a phenomenological study in the methods section. Otherwise, it appears that the study is more of a generic qualitative study. Important characteristics include the focus on lived experience, the meaning/interpretation of the phenomenon by the study participants, steps taken to bracket or to limit researcher bias/ influence in the interpretation fo the data etc

Secondly, the introduction needs to provide more information on the local context of the study. Were trained formal caregivers available in the country? How are foreign caregivers employed and what are the laws or restrictions surrounding the employment of these foreign helpers? Do they receive training prior to employment? What is the current long-term care system in Taiwan?

In the introduction section, add justification on why it is important to understand the perspectives of the stroke survivors. Currently it is only briefly and superficially described. What are the possible benefits of knowing their expectations, needs, challenges related to employing foreign caregivers? What expected policy changes are required?

In the methods section, please include a paragraph to represent the positionality of the researchers in the study. This is important for phenomenological studies and is part of the COREQ criteria. What additional steps were taken to minimise personal bias / influence?

Provide more description on the development of the topic guide. Was it based on theory or specific research questions? Please provide the full topic guide.

What language was used to conduct the interview? What language was used for transcription and analysis?

Line 140-141: How does presence of family caregivers influence the interview?

Lines 152: Provide an explanation for textural descriptions with a reference. It is good to provide a sample of the coding process.

Line 153: usually the term used is "thematic saturation"

A total of 23 stroke survivors were recruited for the study. How did the researchers determine when recruitment would be terminated?

For the results section, phenomenological studies usually provide rich description which can illustrate the lived experience.The current results section seem to be inadequate. For example, under Severity or complexity, the provided quotes are inadequate to reflect the severity or complexity of the patient's condition.

Overall, the results section show a more superficial description of the study findings rather than providing an interpretation of the lived experience of the stroke survivors.

The discussion provided a good explanation of the cultural setting of expectations towards foreign caregivers. The authors could provide some conclusion on the ethics or appropriateness of these expectations. Provide some recommendations on policy or education for potential employers of foreign caregivers.

Please complete the COREQ checklist and upload it as a supplementary information for this study.

Thank you.

6. PLOS authors have the option to publish the peer review history of their article (what does this mean?). If published, this will include your full peer review and any attached files.

Reviewer #1: **Yes: **Åsa Rejnö

Reviewer #2: **Yes: **Chai-Eng Tan

---

## [Author Response · Author response to Decision Letter 0]

4 Nov 2024

Response: We sincerely appreciate the valuable feedback provided by both reviewers, which has greatly improved the quality of our manuscript. We have carefully considered and addressed all of your suggestions and concerns in this revision, and we believe the changes have strengthened the clarity and rigor of our work. Please see our itemized responses to each comment below.

5. Review Comments to the Author

Reviewer #1: Review ”Employing foreign caregivers: exploring perspectives of older stroke survivors – a qualitative study”

Thank you for the opportunity to review this manuscript with, from my point of view, a problematic topic. The authors themselves mention exploitation and I become very badly affected when reading the results. The expectations of the persons affected by stroke widely exceeds what is reasonable to expect. How can anyone consider it to be ok to expect persons to work 24 hours per day as is stated in line 250? This represent a very dehumanized way of looking at other persons. The researchers make a good discussion of this very problematic issue.

1. Abstract

In the section for Results, the themes are given, however, not with the correct names. Make sure to use the correct names from the results or re-write the text in the abstract so it is evident that what is written there is not the actual themes, but a summary of what they tell about.

Response: Thank you for pointing out this inconsistency. We have carefully revised the Results section in the abstract to ensure that the theme titles now accurately align with those used in the main text (see Page 3 / Line 47 – 51).

2. Introduction

A little more about the health care system in Taiwan needs to be described for an international audience to be able to understand the article. Which care services are available for persons who are in need of daily care, but not needing hospital care? What is the cost for such services for the individual person in need?

RESPONSE: We added the following background: “Since 1992, the Taiwanese government has introduced live-in foreign caregivers under the regulation titled “Temporary Measure to Address Shortages of Manpower for Providing Care for Households’ Disabled,” in response to the needs of families with disabled members, addressing the crisis of the local caregiver shortage [12]. Although Taiwan’s Ten-Years Long-Term Care Project (LTC) offers various home-based and community-based services, such as adult daycare and home care provided by Taiwanese care workers [13], many individuals with care needs still prefer to hire foreign caregivers. By 2019, approximately 28% of people with disabilities were cared for by live-in foreign caregivers [14]. In 2021, 225,432 foreign caregivers were employed by families to provide care services, compared to only 74,601 Taiwanese care workers [15]. This gap may be attributed to the advantage of live-in foreign caregivers, who offer 24-hour assistance at an affordable cost (USD 550 to 650, per month)[16, 17].”

(see Page 4 - 5, Line 79 – 90)

3. On p 11, lines 86-87, it is stated that “a” scoping review has highlighted but then two references are given. When using two, the word “a” is not sufficient. However, reference 16 does not seem to be a scooping review.

RESPONSE: Thank you for your comment. We have addressed this by removing the redundant reference and now only cite the appropriate scoping review: Salami B, Duggleby W, Rajani F. The perspective of employers/families and care recipients of migrant live-in caregivers: a scoping review. Health & Social Care in the Community. 2017;25(6):1667-78. (see Page 6, Line 99)

4. On p 11, lines 87-90, a statement is given about existing qualitative investigations, where the scoping review is given as one of the references. Are the authors sure that all studies included in the scoping review are performed with qualitative methods?

RESPONSE: We apologize for the confusion. The cited scoping review included a variety of international studies that explored the experiences of families and live-in caregivers; it was not limited to qualitative research. Our intention was to emphasize that the review identified a lack of research in Asia on care recipients' experiences with foreign caregivers, regardless of whether the studies were qualitative or quantitative (while this review did review many qualitative studies regarding the experiences of families and live-in caregivers). To clarify, we have revised this section as follows: “A scoping review of international literature on the experiences of families and live-in caregivers highlighted the lack of research conducted in Asia regarding care recipients' experiences with the care they receive from foreign caregivers [21]. In fact, there is limited literature exploring the perspectives of individuals with care needs and their families on hiring foreign caregivers. Among these studies, some qualitative investigations have shed light on the complex and interdependent relationships between foreign caregivers and their care recipients [21-23].”

(see Page 6, Line 99 – 105)

5. Method

Setting and participants

It is unclear to me if all patients with a previous diagnosis of stroke are scheduled for outpatient visits at the physical medicine and rehabilitation departments of the included hospitals or if the patients who are scheduled for such visits constitute a special selection in some way? I would not expect that everyone who has got a diagnosis of stroke and live in the catchment area of one of the included hospitals, would still be scheduled for outpatient visits at these departments more than five years after their stroke. This needs to be clarified in order to understand if the available participants in any way constitutes a selected group of patients.

RESPONSE: Since Taiwan’s National Health Insurance (NHI) covers all citizens, the out-of-pocket costs for rehabilitation services are minimal. As a result, patients face little financial burden when continuing outpatient rehabilitation after being discharged from the hospital. Additionally, many hospitals do not impose limits on the duration of outpatient rehabilitation programs, even if providers determine that the program is no longer beneficial for the patient [4]. To avoid confusion, we added the following description in the Study Setting and Recruitment: “Under Taiwan’s National Health Insurance (NHI) system, there are no restrictions on the use of outpatient rehabilitation services for stroke patients, allowing us to recruit patients at various stages of recovery.” (see Page7, Lines 130 – 133)

6. Procedure

On p. 13 the development of the semi-structured interview guide is described. I would suggest to also give the number of questions, not only state that there were “various questions” together with the examples tat are given.

RESPONSE: Thank you for this suggestion. We added more information about the interview guide as following: “This guide included seven major open-ended questions, which were reviewed by an interdisciplinary expert panel, as well as stroke survivors, and caregivers. The full interview guide is available in S2 Table, with examples of key questions provided below…” (see Page 9 – 10, Lines 173 – 176)

7. On p 14, line 140, present tense is used, while past tense should be used, on for example “family caregivers are encouraged”.

RESPONSE: We apologize for the error. We revised this sentence to “Family caregivers were encouraged…” (see Page 10, Lines 185)

8. Data analysis

On p 12, line 102 the study is stated to employ a descriptive phenomenological approach, but in the section for Data analysis, p 14, line 149, it is stated that a thematic analysis was performed. These two statements are not compatible and cannot both be true Please revise this. I cannot find any signs in the text, of this study being a phenomenological study so I would suggest to keep the claim about the thematic analysis and omit the phenomenological claims. This also gives a need to revise the aim. When a phenomenological analysis has not been performed, the results is not “lived experience”.

RESPONSE: Thank you for your feedback. Based on a thorough literature review, the nature of our results, and the reviewer's suggestions, we have revised the study's design from a phenomenological approach to a descriptive qualitative approach. (see Page 7, Line 119)

9. Ethical considerations

I cannot find any information concerning ethics despite the statement about ethical approval and information given to the participants found on p 12. Information about if informed consent was obtained, how data was handled and stored and how the participants’ identities have been protected are missing. Also information is lacking about the relationship between the researchers who recruited the participants and the researcher who conducted the interviews, and the participants. Were the researchers in any caregiving position for the participants? Can it be assumed that any power relations between the researchers and the participants existed so that the participants could feel forced to participate?

RESPONSE: We added the ethical information in the Methods section: “Informed consent was obtained from all participants before the interviews, and they were informed of their right to withdraw from the study at any time.” (see Page 7, Lines 122 - 124); “Data was handled confidentially, with all participant information de-identified using unique identification numbers. These identifiers were stored separately from the study data on a secure, password-protected drive. Only the principal investigator and select research staff had access to the encrypted file linking identification numbers to subject identities. The Office of Human Research was responsible for monitoring and auditing the collected data to ensure compliance with ethical standards.” (see Page 11, Lines 195 – 201); and “To minimize personal biases and influence, several steps were taken. YS and YM, who had no prior relationships or power dynamics with any participants, managed participant recruitment to ensure a diverse sample. Additionally, YS conducted all interviews in Mandarin, maintaining neutrality throughout the data collection process.” (see Page 9, Line 162 – 165).

10. Results

In the method section only patients with a diagnosis of stroke are stated to be participants. I was therefore very surprised to understand from line 181-182 and the very first quotation in the results, that information given by relatives to the participants was included. If relative’s statements are to be part of data, then they also needs to be viewed as participants in the study.

RESPONSE: We apologize for the confusion. Since family caregivers only provided assistance to the participants (e.g., repeating their words when their pronunciation was difficult to understand), they were not considered participants in this study. We addressed this in the Methods section as follows: “Family caregivers were encouraged to accompany participants and provide needed assistance, especially for those with mild aphasia that could affect their communication. Their presence was solely for assistance purposes, and if any interference in the interview process was observed, the family caregivers were asked to leave to minimize their influence on the interview.” (see Page 10, Lines 185 – 189) To avoid confusion, we also removed the caregiver’s quote from the results section.

11. There are multiple problems related to the results, mostly pertaining to the headings of the themes and sub themes.

The sub theme “Severity and complexity”, I do not see nether severity nor complexity in the text that describes what is labelled as this. Please revise.

RESPONSE: We revised the subtheme “Severity and complexity” to “Need for supervision and physical assistance.” (see Page 19, Line 239)

12. I am not very sure about the name of the second theme, line 208. Is the use of “survivor” really good? We can assume that no dead person will have the need of a caregiver of any kind. I suggest to use the word participant or the word stroke instead.

RESPONSE: Thank you for the suggestion. We revised Theme 2 to “The expectations of participants toward foreign caregivers.” (see Page 20, Line 268)

13. I have trouble to understand how anyone hiring a foreign caregiver can view language barrier as an “unexpected consequence”. Is really seen in the interviews or is this a result of a bad heading for the sub theme? Make sure that this section reports on the results, not discuss them. 

RESPONSE: We apologize for any confusion caused by the original headings. We have revised the theme name to 'The challenges of employing a foreign caregiver' and changed the subtheme from 'Language barriers' to 'Communication difficulties.” (see Page 24, Line 326)

14. Discussion has a section dedicated to that. Examples that needs to be removed from the results are for example line 225.

RESPONSE: Thank you for the suggestion. We removed the redundant descriptions. 

15. It is not ok to use valuing word in the results section, these should be reserved for the discussion, see for example first word, line 309.

RESPONSE: We removed the word “Unexpectedly” and made sure no other valuing word in the results section. (see Result section)

16. Discussion

Well-written and problematizing the results in a good way. This is really well needed with this kind of problematic and bothering results showing so little respect for other human beings.

RESPONSE: Thank You.

17. Strength and limitations

Discusses the limitations in a proper way. No strengths in the study is reported which I consider would be of value. There cannot only be limitations in the study, there must also exist some good things. I also do not agree that there would be greater reasons for persons with negative experiences to decline participation, than it would be for persons with positive experiences. Those who are dissatisfied are often not hesitant to tell about it.

RESPONSE: We have added the strengths of the study (see Page 32, Lines 477 – 485). Additionally, we have removed the limitation regarding individuals with negative experiences possibly declining to participate in the study, as per your suggestion (see Page 32, Line 486 – 493).

18. Conclusion

Supported by the results.

RESPONSE: Thank You.

19. References

Reference 19 – the title of the publication have been written in a mysterious way: Approaches QIaRDCAF. Similar problem appear in ref 33.

The journal names are alternately written with initial upper-case and lower-case in each word. Make sure to adhere to the journals guidelines concerning this.

RESPONSE: We have corrected the errors. (See Pages 35 – 36, Ref. [25] and [42])

20. Language

Good

Reviewer #2: Thank you for addressing an important phenomenon in caregiving for stroke patients in an Asian country. In Asian countries, reliance on informal caregivers is common due to the reluctance to institutionalise stroke patients in care homes, related to their local culture of filial piety. The current manuscript still contains many areas that require improvement.

I hope the authors can address the following issues:

1. The study is being presented as a phenomenological study. Please mention the key characteristics of a phenomenological study in the methods section. Otherwise, it appears that the study is more of a generic qualitative study. Important characteristics include the focus on lived experience, the meaning/interpretation of the phenomenon by the study participants, steps taken to bracket or to limit researcher bias/ influence in the interpretation fo the data etc

RESPONSE: Thank you for this critical suggestion. We revised the study design as a descriptive qualitative approach. (see Page 7, Line 119),

2. Secondly, the introduction needs to provide more information on the local context of the study. Were trained formal caregivers available in the country? How are foreign care

---

## [Decision Letter · Decision Letter 1]

12 Nov 2024

PONE-D-24-31334R1Employing foreign caregivers: exploring perspectives of older stroke survivors - a qualitative studyPLOS ONE

Dear Dr. Chang,

Thank you for submitting your manuscript to PLOS ONE. After careful consideration, we feel that it has merit but does not fully meet PLOS ONE’s publication criteria as it currently stands. Therefore, we invite you to submit a revised version of the manuscript that addresses the points raised during the review process.

**ACADEMIC EDITOR: **

The authors have addressed most of the comments from the reviewers satisfactorily. There are only minor corrections required. I suggest that the manuscript be sent for professional proofreading prior to resubmission as PLOS ONE does not copy edit accepted manuscripts. 

We look forward to receiving your revised manuscript.

Kind regards,

Chai-Eng Tan

Academic Editor

PLOS ONE

Journal Requirements:

Reviewers' comments:

Reviewer's Responses to Questions

**Comments to the Author**

1. If the authors have adequately addressed your comments raised in a previous round of review and you feel that this manuscript is now acceptable for publication, you may indicate that here to bypass the “Comments to the Author” section, enter your conflict of interest statement in the “Confidential to Editor” section, and submit your "Accept" recommendation.

Reviewer #1: All comments have been addressed

Reviewer #2: (No Response)

2. Is the manuscript technically sound, and do the data support the conclusions?

Reviewer #1: (No Response)

Reviewer #2: Yes

3. Has the statistical analysis been performed appropriately and rigorously? 

Reviewer #1: (No Response)

Reviewer #2: N/A

4. Have the authors made all data underlying the findings in their manuscript fully available?

Reviewer #1: (No Response)

Reviewer #2: Yes

5. Is the manuscript presented in an intelligible fashion and written in standard English?

Reviewer #1: (No Response)

Reviewer #2: No

6. Review Comments to the Author

Reviewer #1: (No Response)

Reviewer #2: Thank you for addressing most of the comments in the first round of review.

There are still some minor language issues that require revision.

Introduction, line 65: instead of saying "a high number of family caregivers are needed", it would be more appropriate to state that "the caregivers' role is crucial". It is not so much the number of family caregivers, but the availability and quality of care by the caregivers that enables stroke survivors to return to the community safely.

Methods, Design, line 119-120: Please correct the lanugage ".. to understand stroke survivors' experiences in receiving care from..."

Methods, Design, line 124: Please remove the Joanna Briggs Institute Critical Appraisal Checklist from the text. A critical appraisal checklist is meant for evaluating quality of published studies and not to guide reporting. The COREQ checklist is adequate.

Methods, Study Setting and Recruitment, line 132-133: Please describe how you approached potential participants. Were they approached at the waiting area of the rehabilitation services or were they recruited through an advertisement or poster? Who did the recruitment? Were the interviews conducted on the spot or an appointment given for a protected time for the interview?

Study participants, line 140: please replace "Chinese" with "Mandarin" to standardise the manuscript (as mentioned in line 183).

Formatting of quotes, e.g. line 281-283, lines 309-310, lines 343-347, lines 357-358, lines 380-389, lines 394-397. Please separate the quotes from different participants into separate paragraphs.

Discussion, lines 470-473. Please highlight the issue of patient safety when untrained caregivers are used, particularly when one of the quotes mentioned that hte caregiver caused the participant to fall. What does "inadequately treated" mean? Please rephrase for clarity.

References, please ensure references are correctly formatted. Some were incomplete e.g. ref 19, and others used different referencing styles. Referencing for websites should include when the website was accessed.

I suggest that the manuscript undergo professional proofreading prior to resubmission.

7. PLOS authors have the option to publish the peer review history of their article (what does this mean?). If published, this will include your full peer review and any attached files.

Reviewer #1: **Yes: **Åsa Rejnö

Reviewer #2: **Yes: **Chai-Eng Tan

---

## [Author Response · Author response to Decision Letter 1]

10 Dec 2024

Reviewer #1: (No Response)

Reviewer #2: Thank you for addressing most of the comments in the first round of review.

There are still some minor language issues that require revision.

Introduction, line 65: instead of saying "a high number of family caregivers are needed", it would be more appropriate to state that "the caregivers' role is crucial". It is not so much the number of family caregivers, but the availability and quality of care by the caregivers that enables stroke survivors to return to the community safely.

Response: Thank you for the critical feedback. We have revised this sentence to “Family caregivers are essential for reducing this burden and assisting stroke survivors in returning to their homes and communities.” (See page 5, lines 73-75)

Methods, Design, line 119-120: Please correct the lanugage ".. to understand stroke survivors' experiences in receiving care from..."

Response: We have revised this sentence accordingly. (See page 9, lines 145-146)

Methods, Design, line 124: Please remove the Joanna Briggs Institute Critical Appraisal Checklist from the text. A critical appraisal checklist is meant for evaluating quality of published studies and not to guide reporting. The COREQ checklist is adequate.

Response: We have removed the Joanna Briggs Institute Critical Appraisal Checklist and only kept the COREQ checklist. (See page 9, lines 151-152)

Methods, Study Setting and Recruitment, line 132-133: Please describe how you approached potential participants. Were they approached at the waiting area of the rehabilitation services or were they recruited through an advertisement or poster? Who did the recruitment? Were the interviews conducted on the spot or an appointment given for a protected time for the interview?

Response: We described how we approached the potential participants at pages 9-10, lines 161-166): “Patients with stroke were screened for study inclusion eligibility by staff at the rehabilitation clinics, and potentially eligible patients were introduced to the study researchers, who then met with the patients in person and explained the purpose of the study by using informational leaflets.” 

We also described the place where we conducted the interviews at page 13, lines 221-223: The interviews were scheduled at times that were convenient for the participants and took place in quiet settings, either at academic medical centers or in the participants’ homes.

Study participants, line 140: please replace "Chinese" with "Mandarin" to standardise the manuscript (as mentioned in line 183).

Response: We have replaced "Chinese" with "Mandarin" accordingly. (See page 10, line 174)

Formatting of quotes, e.g. line 281-283, lines 309-310, lines 343-347, lines 357-358, lines 380-389, lines 394-397. Please separate the quotes from different participants into separate paragraphs.

Response: We have separated the quotes from different participants into separate paragraphs. (See the Results section pages 22-33) 

Discussion, lines 470-473. Please highlight the issue of patient safety when untrained caregivers are used, particularly when one of the quotes mentioned that hte caregiver caused the participant to fall. What does "inadequately treated" mean? Please rephrase for clarity.

Response: We revised this paragraph and highlighted the issue of patient safety as following: “Training is essential for safety. One participant in the current study recounted an incident in which his caregiver’s inexperience caused him to fall early in their working relationship. Some participants also reported feeling neglected or disrespected. Others did not receive adequate attention from their foreign caregivers.” (See page 30, lines 440-444)

References, please ensure references are correctly formatted. Some were incomplete e.g. ref 19, and others used different referencing styles. Referencing for websites should include when the website was accessed.

Response: Thank you for pointing out this issue. We have thoroughly reviewed all references and ensured their accuracy and consistency with the required formatting style. Additionally, for website references, we have included the date of access as recommended.

I suggest that the manuscript undergo professional proofreading prior to resubmission.

Response: Thank you for your suggestion. We have sent the manuscript for professional proofreading, and the necessary edits have been made to ensure clarity and accuracy.

---

## [Editor Report · Decision Letter 2]

16 Dec 2024

Employing foreign caregivers: A qualitative study of the perspectives of older stroke survivors

PONE-D-24-31334R2

Dear Dr. Chang,

We’re pleased to inform you that your manuscript has been judged scientifically suitable for publication and will be formally accepted for publication once it meets all outstanding technical requirements. Thank you for addressing all the comments posed by the reviewers.

Kind regards,

Chai-Eng Tan

Academic Editor

PLOS ONE
---

## [Editor Report · Acceptance letter]

22 Dec 2024

PONE-D-24-31334R2 

PLOS ONE

Dear Dr. Chang, 

I'm pleased to inform you that your manuscript has been deemed suitable for publication in PLOS ONE. Congratulations! Your manuscript is now being handed over to our production team.

Kind regards, 

on behalf of

Dr. Chai-Eng Tan 

Academic Editor

PLOS ONE